# TreeCaps: Tree-Structured Capsule Networks for Program Source Code Processing

## Abstract

Program comprehension is a fundamental task in software development and maintenance processes. Software developers often need to understand a large amount of existing code before they can develop new features or fix bugs in existing programs. Being able to process programming language code automatically and provide summaries of code functionality accurately can significantly help developers to reduce time spent in code navigation and understanding, and thus increase productivity. Different from natural language articles, source code in programming languages often follows rigid syntactical structures and there can exist dependencies among code elements that are located far away from each other through complex control flows and data flows. Existing studies on tree-based convolutional neural networks (TBCNN) and gated graph neural networks (GGNN) are not able to capture essential semantic dependencies among code elements accurately. In this paper, we propose novel tree-based capsule networks (TreeCaps) and relevant techniques for processing program code in an automated way that encodes code syntactical structures and captures code dependencies more accurately. Based on evaluation on programs written in different programming languages, we show that our TreeCaps-based approach can outperform other approaches in classifying the functionalities of many programs.

## 1 Introduction

Understanding program code is a fundamental step for many software engineering tasks. Software developers often spend more than 50% of their time in navigating through existing code bases and understanding the code before they can implement new features or fix bugs (Xia et al., 2018; Evans Data Corporation, 2019; Britton et al., 2012). If suitable models for programs are built, they can be useful for many tasks, such as classifying the functionality of programs (Nix & Zhang, 2017; Dahl et al., 2013; Pascanu et al., 2015; Rastogi et al., 2013), predicting bugs (Yang et al., 2015; Li et al., 2017; 2018), and providing bases for program translation (Nguyen et al., 2017; Gu et al., 2017).

Different from natural language texts, programming languages have clearly defined grammars and compilable source code must follow rigid syntactical structures and can be unambiguously parsed into syntax trees. There can be complex control flows and data flows among various code elements all over a program that affect the semantic and functionality of the program. Some inter-dependent code elements can appear in an arbitrary order in the program (e.g., a function $A$ calls another function $B$ while $A$ and $B$ are spatially far away from each other); some code elements, such as local variable names, have no significant impact on code functionality.

In the literature, tree-based convolutional neural networks (TBCNNs) have been proposed (Peng et al., 2015; Mou et al., 2016) to show promising results in programming language processing. TBCNNs accept Abstract Syntax Trees (ASTs) of source code as the input, and capture explicit, structural parent-child-sibling relations among code elements. Gated graph neural networks (GGNNs) (Li et al., 2016) are also proposed as a way to learn graphs, and ASTs are extended to graphs with a variety of code dependencies added as edges among tree nodes to model code semantics (Allamanis et al., 2018b). While GGNNs capture more code semantics than TBCNNs, many additional edges among tree nodes have to be added through program analysis techniques, and many of the edges may be noise, contributing longer training time and lower performance. A recent model, known as ASTNN, based on a sequence of small ASTs for statements (instead of one big AST for the whole program) shows better performance than TBCNNs and GGNNs (Zhang et al., 2019).

In this paper, we propose a novel tree-based capsule network architecture, named **TreeCaps**, to capture both syntactical structure and dependency information in code, without the need of explicitly adding dependencies in the trees or splitting a big tree into smaller ones. Capsule Networks (CapsNet) (Sabour et al., 2017) itself is a promising concept that has demonstrated vast potential in outperforming CNNs in various domains including computer vision and natural language processing, because it has the main advantage that it can discover and preserve the relative spatial and hierarchical relations among objects within an input (e.g., an image and a piece of texts).

TreeCaps adapts CapsNet to ASTs for programs, proposes novel primary variable and primary static capsule layers, proposes a novel variable-to-static routing algorithm to route a variable set of capsules to generate a static set of capsules (with the intention of preserving code dependencies), and connects the tree capsule layers to a classification layer to classify the functionality of a program.

In our empirical evaluation, we take various sets of programs written in different programming languages (e.g., Python, Java, C) collected from GitHub and the literature, and train our TreeCaps models to classify programs with different functionalities. Results show that TreeCaps outperforms other approaches in program classification by significant margins, while a study done on variants of the proposed model reveals the effectiveness of the proposed variable-to-static routing algorithm, effect of the dimensionality of the classification capsules on the model performance and the effectiveness of the use of additional capsule layers.

## 2 RELATED WORK

Capsule networks (Sabour et al., 2017; Hinton et al., 2018) use dynamic routing to model spatial and hierarchical relations among objects in an image. The techniques have been successfully applied to different tasks, such as computer vision, character recognition, and text classification (Jayasundara et al., 2019; Rajasegaran et al., 2019; Zhao et al., 2018). None of the studies has considered complex tree data as input. Capsule Graph Neural Network (Zhang & Chen, 2019) has been recently proposed to classify biological and social network graphs, yet, has not been applied to trees for programming languages processing yet.

On the other hand, tree- and graph-based neural networks have been studied for program language processing. TBCNNs (Peng et al., 2015; Mou et al., 2016) have been used to model code syntactical structures. GGNNs (Li et al., 2016; Allamanis et al., 2018b) build dependency graphs from ASTs and use graph neural networks to encode the code dependencies. Variants of TBCNNs and GGNNs are also proposed to represent programs differently and aim to achieve better training accuracy and costs. For example, ASTNN (Zhang et al., 2019) splits an entire AST into a sequence of smaller ones and uses bidirectional gated recurrent units (Bi-GRU) to model the smaller ASTs that represent statements in programs. For another example, bilateral dependency tree-based CNNs (DTBCNNs) (BUI et al., 2019) are used to classify programs across different programming languages. Tree-LSTM (Wei & Li, 2017) has also been used to model tree structures in program code. Our work aims to model tree structures too, but designs special dynamic routing with the intention to capture code dependencies without the need of explicit program analysis techniques.

More generally, there is huge interest in applying deep learning techniques for various software engineering tasks, such as program classification, bug prediction, code clone detection, program refactoring, translation and even code synthesis (Allamanis et al., 2018a; Alon et al., 2019; Hu et al., 2018; Chen et al., 2018; Pradel & Sen, 2018). We are likely the first to adapt capsule networks for program source code processing to capture both syntactical structures and code dependencies, especially for the problem of program classification. In the future, it can be an exciting area to combine more kinds of semantic-aware code representations (e.g., symbolic traces (Henkel et al., 2018)) and tailored program analysis techniques with deep learning to improve code learning tasks.

## 3 APPROACH OVERVIEW

The overview of our TreeCaps approach is summarized in Fig. 1. The source code of the training sample program is parsed into an AST and vectorized with the aid of Word2Vec (Mikolov et al., 2013) or a similar technique that considers the AST node types, instead of concrete tokens, as the vocabulary words (Peng et al., 2015; BUI et al., 2019). The AST and the vectorized nodes are

Figure 1: Approach Overview. The source codes are parsed, vectorized and fed into the proposed TreeCaps network for the program classification task.

then fed in to our TreeCaps network, which consists of a Primary Variable Capsule (PVC) layer to accommodate the varying number of nodes in the AST. Subsequently, the capsules are routed to the Primary Static Capsule (PSC) layer using the proposed variable-to-static routing algorithm, followed by routing with the dynamic routing algorithm to the Code Capsule (CC) layer. Acting as the classification capsule layer, Code capsules capture and provide embeddings for the entire training sample, while denoting the probability of existence of the source code classes by the respective vector norms. Finally, a softmax layer is used on the vector norms to output the probabilities for the input code sample to belong to various functionality classes.

Section 4 and 5 explain more about the AST vectorization and other major components in TreeCaps.

# 4 ABSTRACT SYNTAX TREE VECTORIZATION

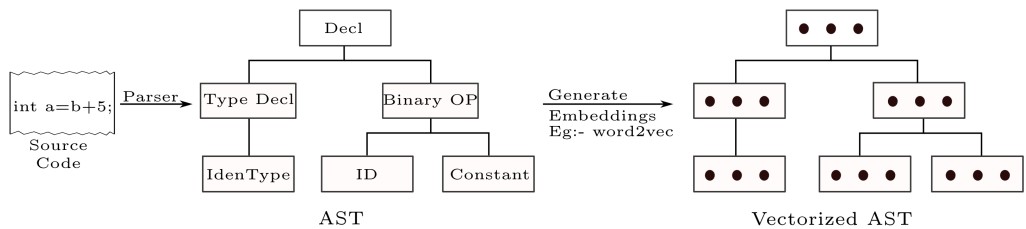

Figure 2: Tree Vectorization, which generates the AST from the source code and vectorizes it using an embedding generation technique.

Fig. 2 illustrates the vectorization of an AST. Every raw source code is parsed with an appropriate parser corresponding to the programming language to generate the AST.[1] The AST represents the syntactic structure of the source code with a set of generalized vocabulary words (i.e., node type names). We use ASTs to train the embedding for node types by applying embedding techniques, such as Word2vec (Mikolov et al., 2013), in the context of ASTs using techniques similar to Peng et al. (2015), which learns a vectorized vocabulary of node types, $\mathbf{x}_{node} \in \mathbb{R}^V$, where $V$ is the embedding size. The learned vocabulary can subsequently be used to vectorize each individual node of the ASTs, generating the vectorized ASTs.

# 5 TREE-BASED CAPSULE NETWORKS

One of the main challenges in creating a tree-based capsule network is that the input of the network is tree-structured (ASTs in our case). Tree-structured data are inherently different from generic image data, $\mathbf{X}_{img} \in \mathbb{R}^{H \times W \times C}$, where $H, W, C$ are the fixed height, width and the number of channels respectively, or natural language data, $\mathbf{X}_{nlp} \in \mathbb{R}^{L \times E}$, where $L, E$ are the fixed padded sentence length and the word embedding size respectively. Hence, the network architecture needs to be constructed to accommodate such tree-structured data, $\mathbf{X}_{tree} \in \mathbb{R}^{T \times V}$, where $T, V$ are the variable tree size (the number of nodes in the tree) and the node embedding size respectively.

## 5.1 PRIMARY VARIABLE TREECAPS LAYER

A further challenge with trees is that the tree size varies from program to program, and the number of children varies from node to node. A naive solution to the problem can be to pad the sizes

---

[1]We use python AST parser for python programs, whereas we use srcML parser for C and Java programs.

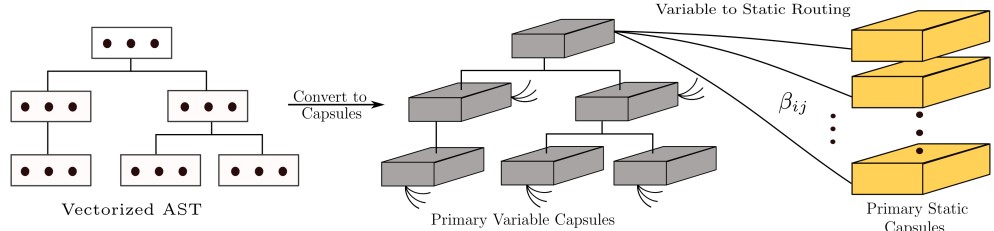

Figure 3: Variable-to-Static Routing, which routes a variable set of capsules to generate a static set of capsules.

to reach a fixed, pre-defined length, following a similar approach in natural language domain to preserve a fixed sentence length. However, zero-padding is not appropriate in our case due to the degree of variability. For instance, the number of children per node can vary from zero to hundreds, causing challenges in deciding the fixed padding length and introducing sparsity. Mou et al. (2016) propose a more effective approach termed as continuous binary tree, where the convolution window is emulated as a binary tree, where the weight matrix corresponding to each node is represented as a weighted sum of three fixed matrices $\mathbf{W}^t, \mathbf{W}^l, \mathbf{W}^r \in \mathbb{R}^{V' \times V}$ and a bias term $\mathbf{b} \in \mathbb{R}^{V'}$, where $V'$ is the embedding size after the convolution, and the weighting coefficients are calculated by taking the positional value in to account.

Hence, for a convolutional window of depth $d$ in the original AST, and there are $K + 1$ nodes (including the parent node) which belong to that window with vectors $[\mathbf{x}_1, ..., \mathbf{x}_{K+1}]$, where $\mathbf{x_i} \in \mathbb{R}^V$, then the convolutional output $\mathbf{y}$ can be defined as follows,

$$\mathbf{y} = tanh(\sum_{i=1}^{K+1} [\eta_i^t \mathbf{W}^t + \eta_i^l \mathbf{W}^l + \eta_i^r \mathbf{W}^r]\mathbf{x}_i + \mathbf{b}) \tag{1}$$

where $\eta_i^t, \eta_i^l, \eta_i^r$ are weights defined corresponding to the depth and the position of the children nodes:

$$\eta_i^t = \frac{d_i - 1}{d - 1} \qquad \eta_i^r = (1 - \eta_i^t)\frac{p_i - 1}{k - 1} \qquad \eta_i^l = (1 - \eta_i^t)(1 - \eta_i^r) \tag{2}$$

where $d_i$ is the depth of the node $i$ in the convolutional windows, $p_i$ is the position of the node and $k$ is the total number of node's siblings. The output of tree-structured convolution resembles the input tree structure, $\mathbf{Y}_{conv} \in \mathbb{R}^{T \times V'}$.

In the Primary Variable Capsule layer, $\mathbf{y}$ obtained from Equation 1 corresponds to the output of one convolutional slice. We use $\varepsilon$ such slices with different random initializations for $\mathbf{W}, \mathbf{b}$, similar to CNNs for image data. Subsequently, as illustrated by Fig 3, we group the convolutional slices together to form $N_{pvc} = \frac{T \times V' \times \varepsilon}{D_{pvc}}$ sets of capsules with outputs $\mathbf{u}_i \in \mathbb{R}^{D_{pvc}}$, $i \in [1, N_{pvc}]$, where $D_{pvc}$ is the dimensions of the capsules (i.e., $D_{pvc}$ is the number of instantiation parameters of the capsules) in the PVC layer. In order to vectorize each capsule output $\mathbf{u}_j$ as $\hat{\mathbf{u}}_j$ (to represent the probability of existence of an entity by the vector length), we subsequently apply a non-linear squash function as follows,

$$\hat{\mathbf{u}}_i = \frac{||\mathbf{u}_i||^2}{||\mathbf{u}_i||^2 + 1} \cdot \frac{\mathbf{u}_i}{||\mathbf{u}_i||^2} \tag{3}$$

where $||\hat{\mathbf{u}}_i||_2 \leq 1$. Hence, the output of the primary variable capsule layer is $\mathbf{X_{pvc}} \in \mathbb{R}^{N_{pvc} \times D_{pvc}}$.

## 5.2 PRIMARY STATIC TREECAPS LAYER

The key issue with passing the outputs of the PVC layer, $\mathbf{X_{pvc}}$, to the Code Capsule layer is that the number of capsules, $N_{pvc}$, is variable from one training example to another. Prior to routing the lower level capsules to a set of higher level capsules, the lower dimensional capsule outputs need to be projected to the higher dimensionality, with the aid of the transformation matrix which learns the part-whole relationship between the lower and the higher level capsules (Sabour et al., 2017). However, a trainable transformation matrix cannot be defined in practice with variable dimensions. Thus, the dynamic routing in the literature (Sabour et al., 2017) cannot be applied between a variable set of capsules and a static set of capsules.

### 5.2.1 VARIABLE-TO-STATIC CAPSULE ROUTING

Therefore, we propose a novel variable-to-static capsule routing algorithm, summarized in Algo. 1.

---

**Algorithm 1** Variable-to-Static Capsule Routing

---

1: **procedure** ROUTING($\hat{\mathbf{u}}_i, r, a, b$)
2: $\quad$ $\hat{\mathbf{U}}_{\mathbf{sorted}} \leftarrow sort([\hat{\mathbf{u}}_1, ..., \hat{\mathbf{u}}_{N_{pvc}}])$
3: $\quad$ Initialize $\mathbf{v}_j : \forall i, j \leq a, \mathbf{v_j} \leftarrow \hat{\mathbf{U}}_{\mathbf{sorted}}[i]$
4: $\quad$ Initialize $\alpha_{ij} : \forall j \in [1, a], \forall i \in [1, b], \alpha_{ij} \leftarrow 0$
5: $\quad$ **for** $r$ iterations **do**
6: $\quad\quad$ $\forall j \in [1, a], \forall i \in [1, b], f_{ij} \leftarrow \hat{\mathbf{u}}_i \cdot \mathbf{v}_j$
7: $\quad\quad$ $\forall j \in [1, a], \forall i \in [1, b], \alpha_{ij} \leftarrow \alpha_{ij} + f_{ij}$
8: $\quad\quad$ $\forall i \in [1, b], \boldsymbol{\beta}_i \leftarrow Softmax(\boldsymbol{\alpha}_i)$
9: $\quad\quad$ $\forall j \in [1, a], \mathbf{s}_j \leftarrow \sum_i \beta_{ij} \hat{\mathbf{u}}_i$
10: $\quad\quad$ $\forall j \in [1, a], \mathbf{v}_j \leftarrow Squash(\mathbf{s}_j)$
11: $\quad$ **return** $\mathbf{v}_j$

---

We initialize the outputs of the Primary Static Capsule layer with the outputs of the $a$ capsules with the highest $L_2$ norms in the PVC layer. Hence, the outputs of the PVC layer, $[\hat{\mathbf{u}}_1, ..., \hat{\mathbf{u}}_{N_{pvc}}]$, are first sorted by their $L_2$ norms, to obtain $\hat{\mathbf{U}}_{\mathbf{sorted}}$, and then the first $a$ vectors of $\hat{\mathbf{U}}_{\mathbf{sorted}}$ are assigned as $\mathbf{v}_j, j \leq a$. The intuition is that, in practice, not every node of the AST contributes towards source code classification. Often, source code consists of non-essential entities, and only a portion of all entities determine the code class. Since the probability of existence of an entity is denoted by the length of the capsule output vector ($L_2$ norm), we only consider the entities with the highest existence probabilities for initialization. It should be noted that the capsules with the $a$-highest norms are used only for initialization, the actual outputs of the primary static capsules are determined by iteratively running the variable-to-static routing algorithm.

A well-known property of source code is that dependency relationships may exist among entities that are not spatially co-located. Therefore, we route $b$ nodes in the AST based on the similarity between them and the primary static capsule layer outputs, where $a \leq b \leq N_{pvc}$. We assign $b = N_{pvc}$ in general to route with all the nodes in the AST. If computational complexity is critical, we can choose a smaller $b$ and route with top-$b$ nodes of the AST. $a$ and $b$ can be chosen empirically, where computational complexity also factors in when choosing $b$.

We initialize the routing coefficients as $\alpha_{ij} = 0$, equally to all the capsules in the primary variable capsule layer. Subsequently, as illustrated by Fig 3, they are iteratively refined based on the agreement between the current primary static capsule layer outputs $\mathbf{v}_j$ and the primary dynamic capsule layer outputs $\hat{\mathbf{u}}_i$. The agreement in this case is measured by the dot product, $f_{ij} \leftarrow \hat{\mathbf{u}}_i \cdot \mathbf{v}_j$, and the routing coefficients are adjusted with $f_{ij}$ accordingly. If a capsule $\gamma$ in the primary dynamic layer has a strong agreement with a capsule $\delta$ in the primary static layer, then $f_{\gamma\delta}$ will be positively large, whereas if there is a strong disagreement, then $f_{\gamma\delta}$ will be negatively large. Subsequently, the sum of vectors $\hat{\mathbf{u}}_i$ is weighted by the updated $\beta_{ij}$ to calculate $\mathbf{s}_j$, which is then squashed to update $\mathbf{v}_j$.

## 5.3 CODE CAPSULE LAYER

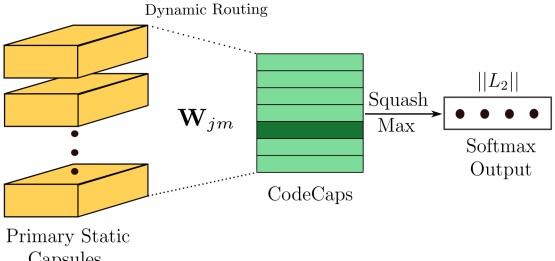

Figure 4: Dynamic Routing between the Primary Static Capsules and the Code Capsules.

Code Capsule layer is the final layer of the TreeCaps network, which acts as the classification capsule layer, as illustrated by Figure 4. Since the outputs of the PSC layer $\mathbf{X}_{\mathbf{psc}} \in \mathbb{R}^{N_{psc} \times D_{psc}}$, where $N_{psc} = a$ and $D_{psc} = D_{pvc}$, consist of a fixed set of capsules, it can be routed to the CC layer via the dynamic routing algorithm in the literature (Sabour et al., 2017) (summarized in Algo. 2). For

each capsule $j$ in the PSC layer, and for each capsule $m$ in the CC layer, we multiply the output of the primary static capsule $\mathbf{v}_j$ by the transformation matrices $\mathbf{W}_{jm}$ to produce the prediction vectors $\hat{\mathbf{v}}_{m|j} = \mathbf{W}_{jm}\mathbf{v}_j$. The trainable transformation matrices learn the part-whole relationships between the primary static capsules and the code capsules, while effectively transforming $\mathbf{v}_j$'s into the same dimensionality as $\mathbf{z}_m$, where $\mathbf{z}_m$'s denote the outputs of the code capsule layer. Similar to the variable-to-static capsule routing, we initialize the routing coefficients $\gamma_{jm}$ equally, and iteratively refine them based on the agreements between the prediction vectors $\hat{\mathbf{v}}_{m|j}$ and the code capsule outputs $\mathbf{z}_m$, where $\mathbf{z}_m = squash(\sum_j \gamma_{jm} \hat{\mathbf{v}}_{m|j})$.

---

**Algorithm 2** Dynamic Routing

---

1: **procedure** ROUTING($\hat{\mathbf{v}}_j, t, a, c$)
2:      Initialize $\forall j \in [1, a], \forall m \in [1, c], \delta_{jm} \leftarrow 0$
3:      **for** t iterations **do**
4:          $\forall j \in [1, a], \gamma_j \leftarrow softmax(\delta_j)$
5:          $\forall m \in [1, c], \mathbf{z}_m \leftarrow squash(\sum_j \gamma_{jm} \hat{\mathbf{v}}_{m|j})$
6:          $\forall j \in [1, a], \forall m \in [1, c], \delta_{jm} \leftarrow \delta_{jm} + \hat{\mathbf{v}}_{m|j} \cdot \mathbf{z}_m$
7:      **return** $\hat{\mathbf{z}}_m$

---

The primary static capsule outputs are routed to the CC layer using the dynamic routing algorithm, as illustrated by Fig 4, to produce the final capsule outputs $\mathbf{X_{cc}} \in \mathbb{R}^{N_{cc} \times D_{cc}}$, where $N_{cc} = \kappa$ is the number of classes and $D_{cc}$ is the dimensionality of the code capsules. Ultimately, we calculate the probability of existence of each class by obtaining $L_2$ norm of each CodeCaps output vector.

### 5.4 MARGIN LOSS FOR TREECAPS TRAINING

We use the Margin Loss proposed by Sabour et al. (2017) as the loss function for TreeCaps. For every code capsule $\mu$, the margin loss $L_\mu$ is defined as follows,

$$L_\mu = T_\mu \max(0, m^+ - \|v_\mu\|)^2 + \lambda(1 - T_\mu) \max(0, \|v_\mu\| - m^-)^2 \tag{4}$$

where $T_\mu$ is 1 if the correct class is $\mu$ and zero otherwise. Following Sabour et al. (2017), $\lambda$ is set to 0.5 to control the initial learning from shrinking the length of the output vectors of all the code capsules, and $m^+, m^-$ are set to $0.9, 0.1$ as the lower bound for the correct class and the upper bound for the incorrect class respectively.

## 6 EMPIRICAL EVALUATION

### 6.1 DATASETS AND IMPLEMENTATION

We used three datasets in three programming languages to ensure cross-language robustness. The first dataset (**A**) contains 6 classes of sorting algorithms, with 346 training programs on average per class, written in Python.[2] The second dataset (**B**) is inherited from BUI et al. (2019), which contains 10 classes of sorting algorithms, with 64 training programs on average per class, written in Java. The third dataset (**C**) is inherited from Mou et al. (2016), which contains 104 classes of C programs, with 375 training programs on average per class. For the dataset A, we used the publicly available vectorizer (see Footnote 2) to generate embeddings for more than 90 AST node types in Python. For the datasets B & C, srcML[3] defines a unified vocabulary for more than four hundred AST node types for C and Java (and several other languages, but not Python), and we adapted the same vectorizer to generate embeddings for the unified AST node types defined by srcML.

We used Keras and Tensorflow libraries to implement TreeCaps. To train the models, we used the RAdam optimizer (Liu et al., 2019) with an initial learning rate of 0.001 subjected to decay, on an Nvidia Tesla P100 GPU. To enhance the classification accuracies, a weighted average ensembling technique (Krogh & Vedelsby, 1995) was used.

Table 1: Comparison of TreeCaps with other approaches. The means and the standard deviations from 3 trials are shown.

| Model | Dataset A | Dataset B | Dataset C |
|---|---|---|---|
| GGNN  Allamanis et al. (2018b) | - | 85.00% | 86.52% |
| TBCNN (Mou et al., 2016) | 99.30% | 75.00% | 79.40% |
| TreeCaps | 100.00 ± 0.00% | 92.11 ± 0.90% | 87.95 ± 0.23% |
| TreeCaps (3-ensembles) | **100.00%** | **94.08%** | **89.41%** |

## 6.2 PROGRAM CLASSIFICATION RESULTS

Table 1 compares our results to other approaches for program classification. It should be noted that, Mou et al. (2016) have used custom-trained initial embeddings for a small set of about 50 AST node types defined specifically for C language only (Peng et al., 2015) and reported a higher result in their paper, while our approach generates the initial embeddings for a much larger vocabulary of more than three hundred unified AST node types for both C and Java. For a fairer comparison based on the same set of AST node vocabulary, especially for the datasets B & C, we used our embeddings based on srcML node vocabulary as the initial embeddings across all models. We followed the techniques proposed in Allamanis et al. (2018a) and BUI et al. (2019) to re-generate the results for GGNN and the techniques proposed in Mou et al. (2016) to re-generate the results for TBCNN.

For the dataset A, we achieved a perfect classification result, outperforming TBCNN by a narrow margin of less than 1%. However, the margin was more significant for the datasets B and C. An average accuracy of 92.11% was achieved by our approach for the dataset B, outperforming other approaches at least by 7.11%. TreeCaps outperformed its convolutional counterpart (TBCNN) by a significant margin of 17.11%. The performance was further improved by 1.97% with the use of 3-model weighted average ensembling technique. For the dataset C, our approach was able to surpass the other approaches by 1.43%, achieving an accuracy of 87.95%. However, TreeCaps surpassed TBCNN by a more significant margin of 8.55%. Three-model weighted average ensembling on the dataset C provided a further improvement of 2.89% in comparison to the other approaches, achieving an accuracy of 89.41%.

## 6.3 MODEL ANALYSIS

We evaluate the effects of various aspects of the TreeCaps model, including the effect of the variable-to-static routing algorithm, variations in the number of instantiation parameters in the CodeCaps layer, and the addition of a secondary capsule layer. We evaluate these variations on the dataset B.

Table 2: Effect of different model variants

| Model Variant | Accuracy |
|---|---|
| Variable-to-Static Routing Algorithm $\rightarrow$ Dynamic Pooling | 83.43% |
| Instantiation parameters $\rightarrow D_{cc} = 4$ | 90.90% |
| $D_{cc} = 8$ | 92.10% |
| $D_{cc} = 12$ | 90.33% |
| $D_{cc} = 16$ | 91.51% |
| TreeCaps $\rightarrow$ TreeCaps + Secondary Capsule Layer | 92.31% |
| TreeCaps with Variable-to-Static Routing and $D_{cc} = 8$ | 92.11% |

### 6.3.1 EFFECT OF THE VARIABLE-TO-STATIC ROUTING ALGORITHM

We investigate the effect of the variable-to-static routing algorithm by replacing it with Dynamic Max Pooling (DMP). Since there is no alternative approach existing in the literature for routing a variable set of capsules to a static set of capsules, we compare the proposed routing algorithm with dynamic pooling. The output of the PVC layer, $\mathbf{X_{pvc}} \in \mathbb{R}^{N_{pvc} \times D_{pvc}}$ consists of a variable component, $N_{pvc}$. Using dynamic max pooling across all the $N_{pvc}$ capsules will result in one output capsule, $\mathbf{X_{dmp}} \in \mathbb{R}^{1 \times D_{pvc}}$. Since $\mathbf{X_{dmp}}$ has no variable components across the training samples,

---

[2]Collected from https://github.com/crestonbunch/tbcnn.

[3]https://www.srcml.org/

it can now be routed to the code capsules using the dynamic routing algorithm. However, it should be noted that DMP is not suitable for capsule networks, as it destroys the spatial and dependency relationships between the capsules. We use DMP here only for comparison purposes.

As summarized in Table 2, DMP yields a considerably lower accuracy of $83.43\%$ than our routing algorithm by a significant margin of $8.68\%$, establishing the effectiveness of our proposed algorithm.

### 6.3.2 EFFECT OF THE NUMBER OF INSTANTIATION PARAMETERS

The instantiation parameters $D_{cc}$ of the Code Capsule layer acts as the final embeddings used for classification, in other words, the dimensionality of the latent representation of source code. If the dimensionality of the latent representation is higher than required, it can introduce sparsity and/or correlations between the instantiation parameters, reducing the classification accuracy. On the contrary, if the dimensionality of the latent representation is too low, it may not be sufficient to capture the variations in source code, leading to under-representation, reducing the classification accuracy. Hence, in an attempt to identify a suitable value for $D_{cc}$ for source code classification, we investigate the effect of $D_{cc}$ in the accuracy. As summarized in Table 2, we observed that the most suitable value was $D_{cc} = 8$ for the dataset B.

### 6.3.3 EFFECT OF THE ADDITION OF A SECONDARY CAPSULE LAYER

We evaluate the addition of an extra capsule layer functionally similar to a primary static capsule layer, which we call the secondary capsule (SC) layer. With respect to Fig 1, we stack the SC layer in between the PSC layer and the CC layer. We use dynamic routing to route between the PSC and SC layers and between the SC and CC layers. Even though we observed a minor improvement of the classification accuracy, the added computational complexity increases the inference time by $16\%$, from $10.3\ ms$ to $11.9\ ms$ per sample. The usefulness of the addition of such a SC layer needs to be further investigated.

### 6.4 DISCUSSION OF LIMITATIONS & FUTURE WORK

Since TreeCaps is based on the capsule networks, it inherits the limitations of capsule networks such as the high computational complexity in comparison to CNNs, and performance reduction with the increasing number of classes. TreeCaps lacks a reconstruction network, similar to the reconstruction network for image data presented by Sabour et al. (2017), which is useful to investigate the interpretability of the capsule network, including the relationship between the learnt instantiation parameters and the physical attributes of data.

We intend to extend our work to further investigate the effects of different initial embeddings on the classification accuracy. Further, we intend to compare related pieces of code identified by TreeCaps to program dependencies identified by program analysis techniques, to evaluate the effectiveness of TreeCaps as an embedding generating technique, and to extend TreeCaps to other related tasks such as bug detection and localization.

## 7 CONCLUSION

In this paper, we proposed a novel tree-based capsule network (**TreeCaps**) to learn rich syntactical structures and semantic dependencies in program source code. The model proposed novel technical features that deal with variable sized trees across different programming languages, including primary variable and primary static capsule layers, and the variable to static routing algorithm. Our empirical evaluations show that these features significantly contribute to the high classification accuracy of TreeCaps model for program classification tasks for various programming languages. To the best of our knowledge, our work is the first to adapt capsule networks to trees and apply them to program source code learning. We believe that TreeCaps can capture more code semantics than previous code learning models and complement existing program analysis techniques well.

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
