# OpenReview forum: "TreeCaps: Tree-Structured Capsule Networks for Program Source Code Processing"
_ICLR.cc/2020/Conference — Reject_

### Official Review · AnonReviewer3 · 2019-10-20
**Official Blind Review #3**

**Rating:** 3

**Review:**

This paper proposes a tree-structured capsule network for program source code processing (essentially a program classification task with three datasets).

The idea of incorporating tree structures into the design for capsule networks is not wrong. However, I am not sure why a capsule network is even needed in program classification. The authors follow the cliché of the importance of tree structures, but show little insight into the use of capsule networks in program analysis. The authors started rationalizing the capsule networks by saying "Capsule Networks itself is a promising concept ..." Being a promising concept itself doesn't necessarily mean it has/is suitable to be applied in program classification.

The treatment in Sec. 5.1 of the tree structures is pretty the same as in Mou et al. [2016], linearly weighting a token by its position. Sec. 5.2 is extremely hard to understand. It starts with presenting an algorithm and its line-by-line interpretation. I know how to program, but I wish to get some intuition of why capsule networks are needed for program classification, and how it is different from a generic capsule network and/or a graph capsule network [Xinyi & Chen, 2019]. Given a graph capsule network is in place, I found the contribution of this paper (tree capsule network) is limited.

The experiments are very thin. The authors only compare their results to TreeCNN and Gated  Graph NN (GGNN). It's unclear if TreeCaps is better than other existing models, such as Transformer, TreeTransformer, GraphCap, etc.

While the authors experimented on three datasets, the evidence is actually limited. Dataset A is saturated (99.3%--100%). Dataset B shows some performance improvement (compared with GGNN and TreeCNN only). Dataset C basically shows TreeCaps is similar to GGNN. The gap between 89.41% and 86.52% is largely due to model ensembles. But the performance of GGNN ensembles is unknown.

In summary, the paper applies Capsule Network to tree structures. The authors mainly follow the cliché of tree structures, but are not too excited about the capsule stuff. I am not excited either.

==
Minor:

Nghi D. Q. BUI -> misformatted. Probably they are two people.
Zhang Xinyi and Lihui Chen --> Not sure if Xinyi is the last name.

**Experience Assessment:**

I have published in this field for several years.

**Review Assessment: Checking Correctness Of Derivations And Theory:**

N/A

**Review Assessment: Checking Correctness Of Experiments:**

I carefully checked the experiments.

**Review Assessment: Thoroughness In Paper Reading:**

I read the paper thoroughly.

---

> ### Author Response · Authors · 2019-11-15
> **Response to Reviewer #3**
>
> We would like to thank the reviewer for his valuable time, helpful feedback and insightful suggestions to further improve our study.
>
> Q3-1: Intuition behind the use of Capsule Networks for Program Classification
>
> Response:
>
> We thank the reviewer for this very important comment.  We kindly invite the reviewer to refer to the common response above. (https://openreview.net/forum?id=SJgXs1HtwH&noteId=r1eiYk0oiH)
>
> Q3-2: Empirical Results
>
> Response:
>
> We acknowledge the reviewer’s concern with respect to the experimental results. We plan to conduct additional experiments and compare TreeCaps performance with other existing approaches such as Transformer, TreeTransformer networks, GraphCap, and etc. Further, we intend to conduct experiments to establish the robustness of TreeCaps with other large datasets in our future studies.
>
> We acknowledge that the performance gain between TreeCaps and GGNN for Dataset C is not very high. Yet, without ensembling, TreeCaps still achieve a 1.43% gain over GGNN, which we believe to be significant in accordance with the currently accepted standards in the community. Each result shown consist of the mean and the standard deviation of 3 independent trials with random initialization.
>
> Q3-3: Minor revisions
>
> Response:
>
> We thank the reviewer for pointing out these issues.
>
> Respectfully, we believe that Nghi D. Q. BUI is one person.
>
> For Xinyi Zhang, the reviewer is right that Zhang should be the family name.

---

### Official Review · AnonReviewer1 · 2019-10-23
**Official Blind Review #1**

**Rating:** 1

**Review:**

The paper proposes a capsule-network-based architecture for predicting program properties and is evaluated on three tasks for predicting an algorithm from a code snippet.

Technically, the paper aims to transfer the idea of convolution from images and apply it to abstract syntax trees of programs. To do this, two dimensions describing the position of a node in a tree position are used - the depth of a node in a tree and its index in the list of children of its parent. This choice, however, is similar to image convolutions only at a very artificial level and drops significant amount of semantically-interesting information for programs from the index of the node at the parents, while keeping the total depth (which rarely matters in programs, as code is usually semantically similar no matter how nested in other code it is).

The experiments are small (on two small and one slightly larger dataset) and inconclusive:
1) Given the number of experiments done for tuning parameters on Dataset B (with ~640 examples), it is not clear that we are not observing some trivial case of overfitting. The improvement over GGNN is quite small and mostly due to ensembles.
2) The problem of small evaluation datasets make the results inconclusive. Only Dataset C is sufficiently large, if I assume no optimization like for Dataset B was performed.
3) Furthermore, it looks like the considered tasks are may be better handled by models such as code2vec or code2seq than by GGNN. The paper needs to include stronger baselines.


**Experience Assessment:**

I have published one or two papers in this area.

**Review Assessment: Checking Correctness Of Derivations And Theory:**

I carefully checked the derivations and theory.

**Review Assessment: Checking Correctness Of Experiments:**

I carefully checked the experiments.

**Review Assessment: Thoroughness In Paper Reading:**

I read the paper at least twice and used my best judgement in assessing the paper.

---

> ### Author Response · Authors · 2019-11-15
> **Response to Reviewer #1**
>
> We would like to thank the reviewer for his valuable time, helpful feedback and insightful suggestions to further improve our study.
>
> Q1-1: Convolutions may drop significant amount of semantically-interesting information
>
> Response:
>
> We acknowledge the reviewer’s concern that the Tree-convolutions may drop significant amount of semantically-interesting information. We adopted the approach proposed by Mou et al. (2016), since we believe to the best of our knowledge that it is the most effective tree-convolution technique in the literature. Improving the tree-convloution technique was not in the scope of this study. However, we agree that a careful modification to the existing approach, or investigating a novel approach to preserve semantically interesting information in tree-convolution would certainly be a very interesting study, and would improve the performance of tree-convolution based networks.
>
> On the other hand, we hypothesis that TreeCaps can still learn relevant useful dependency/semantic relationships among program elements that are not spatially co-located while the network is training, without explicitly providing additional information or constraints.
>
> Q1-2: Experiment Results
>
> Response:
>
> We acknowledge the reviewer’s concern with respect to the experimental results.
>
> 1) The results presented in Section 6.3 were intended as an ablation study, to demonstrate the effects of different aspects of TreeCaps such as the proposed variable to static algorithm and the dimensionality of the classification capsule output. Apart from the experiments with varying dimensionalities of the code capsule output (which was intended as a demonstration of under or redundant latent representation), we did not conduct any dataset-specific hyperparameter tuning to improve the performance. Each result shown consist of the mean and the standard deviation of 3 independent trials with random initialization. Thus, we do not believe that the gains are due to a trivial case of overfitting.
>
> 2) As the reviewer has presumed correctly, we did not conduct any optimization for Dataset C. We used these datasets due to the limited availability of suitable datasets. (with respect to resource constraints, etc.) As the reviewer has kindly suggested, we intend to conduct further experiments to establish the robustness of TreeCaps with other large datasets in our future studies.
>
> 3) We plan to conduct additional experiments and compare TreeCaps performance with other existing approaches such as code2vec and code2seq.

---

### Official Review · AnonReviewer2 · 2019-10-24
**Official Blind Review #2**

**Rating:** 1

**Review:**

The paper proposes a neural architecture for summarizing trees inspired by capsule networks from computer vision. The authors re-use a tree convolution from previous work for the bottommost layer, and then propose adaptations to the dynamic routing from capsule networks so that it can be applied to variable-sized trees. The paper applies the proposed architecture to three different program classification datasets, which are in three different languages. The paper reports empirical gains compared to two architectures proposed by previous work.

I think that it's interesting to apply the capsule network architecture to tree classification, but unfortunately it doesn't appear that some of the motivation for capsule networks on images didn't seem to transfer neatly to this setting; for example, there is no equivalent of inverse graphics as there is no reconstruction loss (as pointed out by the authors in Section 6.4).

Also, the variable-to-static capsule routing indeed appears novel, but I was a bit confused by its internal details. It appears that the outputs of the previous layer which occur most often will get routed (considering lines 6-8 of Algorithm 1 which up-weights each of the $\hat{u}_i$ based on its similarity to $v_j$; the $v_j$ are initially a re-numbered subset of $\hat{u}_i$), without any prior transformation of the previous layer first. It seems to me that this doesn't allow for the prior layer to predict more complex features about the input that the subsequent layer is expected to capture. In fact, for certain code classification tasks, it may be that rare capsule outputs from the initial layer are the most important to preserve.

My biggest concern has to do with the empirical results. The source of Dataset C (Mou et al 2016, https://arxiv.org/pdf/1409.5718.pdf) reports 94.0% accuracy in Table 3 on their TBCNN method on the same dataset, whereas this paper reports 79.40% accuracy for TBCNN. I understand that the later result comes from a reimplementation, but it seems fairer to compare against (or additionally report) the results from the original authors of the method.

Also, the paper cites ASTNN (Zhang et al 2019, https://dl.acm.org/citation.cfm?id=3339604) in the introduction, and even though that paper reports (in table 2) 98.2% accuracy on Dataset C, the results table of the paper under review does not mention this in the evaluation section. I don't think that a paper necessarily has to achieve empirical results beating all previous ones in order to merit acceptance, but the way that the comparison is currently set up doesn't seem to facilitate a clear comparison of the pros and cons of this method versus other ones in the literature.

For the above reasons, I vote to reject the paper. For future submissions, it would be good to see a more comprehensive empirical comparison of the proposed method compared to others, and also to have more explanations about the design of the network.

**Experience Assessment:**

I have read many papers in this area.

**Review Assessment: Checking Correctness Of Derivations And Theory:**

I assessed the sensibility of the derivations and theory.

**Review Assessment: Checking Correctness Of Experiments:**

I assessed the sensibility of the experiments.

**Review Assessment: Thoroughness In Paper Reading:**

I read the paper thoroughly.

---

> ### Author Response · Authors · 2019-11-15
> **Response to Reviewer #2**
>
> We would like to thank the reviewer for his valuable time, helpful feedback and insightful suggestions to further improve our study.
>
> Q2-1: It doesn’t appear that some of the motivation for capsule networks on images didn’t seem to transfer neatly to this setting;  for example, there is no equivalent of inverse graphics as there is no reconstruction loss.
>
> Response:
>
> We thank the reviewer for this very important comment.  We kindly invite the reviewer to refer to the common response above. (https://openreview.net/forum?id=SJgXs1HtwH&noteId=r1eiYk0oiH)
>
> Q2-2: Variable to Static Routing Algorithm
>
> Response:
>
> We acknowledge the reviewer’s concern with respect to the preservation of rare capsules. The initialization of $v\_j$ is based on the length of the capsule output vector (L2 norm), which represents the probability of existence of the entity learnt by the capsule. For a given training/testing sample, not all the capsules are activated in a given layer, and source code often consists of non-essential entities, where only a portion of all entities determine the code class. Hence, we initialize the next layer with the capsules which represent entities with highest probability of existence (in other words, highest activation), and dynamically route the rest of the capsules based on the similarity between the respective vector outputs. Therefore, it is not necessarily the capsules that occur most often that get routed to the next layer, instead it is the capsules with the most prominent outputs along with the capsules with the highest vector similarities to them. In this way, rare capsules are still preserved and routed to the next layer.
>
> Further, we acknowledge that we do not use prior transformations between the primary dynamic layer and the primary static layer. However, in the layers subsequent to the primary static layer, we use prior transformations aiding them to predict more complex features. We can use multiple layers similar to the code capsule layer to predict further complex features, depending on the complexity of the classification task. According to Section 6.3.3, empirical evidence suggests that using more such layers is not very effective for the three particular datasets that we have used. However, more complex datasets may benefit from stacking multiple code capsule layers.
>
> Q2-3: Empirical Results
>
> Response:
>
> We acknowledge the reviewer’s concern with respect to the empirical results. The primary reason for the ambiguity between TBCNN [Mou et al. (2016)] and our re-implementation is the initial embeddings, as explained in Section 6.2. Mou et al. (2016) have used custom-trained initial embeddings for a small set of about 50 AST node types defined specifically for C language only, while our approach generates the initial embeddings for a much larger vocabulary of more than three hundred unified AST node types for both C and Java. We decided to follow a more generalized approach across programming languages, at the expense of performance gain resulting from small, specific vocabularies.
> We believed that it would be more general and fairer to compare across datasets in more than one programming language by using the same (and larger) set of AST node vocabulary used in our approach.
>
> We acknowledge the reviewer's perspective on the fairness of the results and potential errors or discrepancies in our re-implementation of TBCNN [Mou et al. (2016)]. Retrospectively, in addition to using the larger set of AST node vocabulary, we should have also applied our approach directly to the initial embeddings with the same smaller set of AST node vocabulary used in TBCNN [Mou et al. (2016)] and ASTNN [Zhang et al. (2019)] etc. for the dataset in C language so that we may have a clearer comparison.

---

### Author Response · Authors · 2019-11-15
**Response to Reviewer#2 Q#1 and Reviewer#3 Q#1: Intuition behind the use of Capsule Networks for Program Classification**

We would like to thank the reviewers for their valuable time, helpful feedback and insightful suggestions to further improve our study.

Q2-1: It doesn’t appear that some of the motivation for capsule networks on images didn’t seem to transfer neatly to this setting;  for example, there is no equivalent of inverse graphics as there is no reconstruction loss.

Q3-1: Intuition behind the use of Capsule Networks for Program Classification

Response:

Among  others,  the  primary  motivation  behind  the  use  of  capsule  networks  for  program  source code classification is the hypothesis that they automatically learn dependency relationships existing  among  entities  that  are  not  spatially  co-located,  due  to  the  proposed  variable  to  static  routing. It is widely accepted that dependency information can greatly aid program source code related tasks. Most graph networks need the dependency information to be externally integrated [BUI et al.(2019)]. Even GraphCaps [Zhang & Chen (2019)] does not address the dependency relationships in their study.

Variable to static routing recognizes the capsules representing the entities with the highest probability of existence, and routes the capsules which have similar vector outputs to them.  As a result,capsules representing entities with dependencies will be routed together, and in the subsequent layers, this dependency information can be utilized for prediction. Hence, we hypothesis that TreeCaps learn the relevant useful dependency relationships while the network is training, without explicitly providing additional information or constraints.

However, we acknowledge that we require additional experiments not included in the manuscript to justify the hypothesis with respect to the dependency relationships (whether TreeCaps learns the dependencies among entities as expected), despite the performance gain of TreeCaps in comparison to a few other existing approaches. We are currently conducting studies to justify this hypothesis, and we summarize the procedure as follows.  We integrate a back-tracking mechanism after a forward pass with a given test case, which identifies the primary variable capsules with k-highest coupling coefficients, connected to a given primary static capsule.  We then trace the entities in the source code corresponding to the identified primary variable capsules and consider them as the entities with dependency relationships as identified by the TreeCaps network. We subsequently compare related pieces of code identified by TreeCaps to program dependencies identified by program analysis techniques to validate our hypothesis.

We acknowledge that we have not used reconstruction loss in this study and thank the reviewer for the kind suggestion.  We believe that reconstruction loss does not enforce the inverse graphics concept alone, instead, it functions as a regularizer which boosts the routing performance by enhancing the pose encoding (in the case of images).  Existing studies on CapsNets for text classification does not use reconstruction loss, yet, manage to capture child-parent relationships well [Zhao et al.(2018)].  However, we agree that the use of a reconstruction loss would have certainly boosted the performance, and aided the learning of dependency relationships. We plan to add the reconstruction loss to TreeCaps, as we mentioned in Section 6.4.

---

### Decision · Program_Chairs · 2019-12-19

**Decision:**

Reject

**Comment:**

This paper proposes an application of capsule networks to code modeling.

I see the potential in this approach, but as the reviewers pointed out, in the current draft there are significant issues with respect to both clarity of motivating the work, and in the empirical results (which start at a much lower baseline than previous work). I am not recommending acceptance at this time, but would encourage the reviewers to clarify the issues raised in the reviews for future submission.